ecology

MCEs, reef connectivity, deep reef refuge hypothesis, benthic community composition, Bermuda

**Author for correspondence:**
Paris V. Stefanoudis
e-mail: paris@nektonmission.org

# Low connectivity between shallow, mesophotic and rariphotic zone benthos

Paris V. Stefanoudis[1,2], Molly Rivers[1], Struan R. Smith[3], Craig W. Schneider[4], Daniel Wagner[5], Helen Ford[1,6], Alex D. Rogers[1,2,†] and Lucy C. Woodall[1,2,†]

[1]Nekton Foundation, Begbroke Science Park, Begbroke Hill, Woodstock Road, Begbroke, Oxfordshire OX5 1PF, UK
[2]Department of Zoology, University of Oxford, Zoology Research and Administration Building, 11a Mansfield Road, Oxford OX1 3SZ, UK
[3]Natural History Museum, Bermuda Aquarium, Museum and Zoo, 40 North Shore Road, Hamilton Parish FL04, Bermuda
[4]Department of Biology, Trinity College, Hartford, CT 06106, USA
[5]NOAA Office of Ocean Exploration and Research, 331 Fort, Johnston Road, Charleston, SC 29412, USA
[6]School of Ocean Sciences, Bangor University, Menai Bridge, Anglesey LL59 5AB, UK

PVS, 0000-0002-4040-8364; SRS, 0000-0002-3078-0015;
CWS, 0000-0003-0506-3791; LCW, 0000-0001-7295-7184

Worldwide coral reefs face catastrophic damage due to a series of anthropogenic stressors. Investigating how coral reefs ecosystems are connected, in particular across depth, will help us understand if deeper reefs harbour distinct communities. Here, we explore changes in benthic community structure across 15–300 m depths using technical divers and submersibles around Bermuda. We report high levels of floral and faunal differentiation across depth, with distinct assemblages occupying each depth surveyed, except 200–300 m, corresponding to the lower rariphotic zone. Community turnover was highest at the boundary depths of mesophotic coral ecosystems (30–150 m) driven largely by taxonomic turnover and to a lesser degree by ordered species loss (nestedness). Our work highlights the biologically unique nature of benthic communities in the mesophotic and rariphotic zones, and their limited connectivity to shallow reefs, thus emphasizing the need to manage and protect deeper reefs as distinct entities.

## 1. Background

Globally, shallow (less than 30 m) coral reef habitats are degrading rapidly as a result of escalating anthropogenic pressures including

†Supervising authors.

overfishing, pollution, coastal development, non-native species invasions and the effects of climate change (i.e. rising seawater temperatures and ocean acidification) [1,2]. By contrast, adjacent, deeper mesophotic (low light) coral ecosystems (MCEs; approx. 30–150 m [3]) are considered to have more stable environmental conditions and subjected to fewer human impacts. As a result, MCEs have been proposed to provide spatial refuge for species common in perturbed shallow-water reefs, which could provide coral propagules to recolonize shallow reefs following disturbance events (a.k.a. the deep reef refuge hypothesis—DRRH; [4,5]), although scientific consensus has yet to be reached (e.g. [6]). For example, even in cases where high faunal connectivity occurs, reduced reproductive performance of MCE inhabitants casts doubt on their ability to replenish shallow reefs [7,8]. Clearly, much more work is required in this sparsely studied environment before attempting any regional or global-scale extrapolations.

Much less is known about faunal connectivity between MCE and deeper reef habitats (greater than 150 m), known as the rariphotic zone (approx. 150–300 m; *sensu* Baldwin *et al*. [9]). This zone was first identified in Curaçao in the Caribbean Sea on the basis of taxonomically distinct fish assemblages dominated by higher occurrences of taxa (e.g. genera, families) that are typically associated with shallow reefs and MCEs rather than deep-water environments [9]. However, whether a rariphotic zone for benthic communities exists has yet to be investigated.

As commercial exploitation of the oceans (e.g. fishing) moves gradually into greater depths [10], and the effects of climate change begin to manifest in MCEs and deeper waters [11–13], the potential of MCE and deeper habitats to act as a refuge for some shallow-water taxa is as timely as ever [14]. This means focusing on vertical connectivity between shallow and deeper reef communities. What is more, a good knowledge of the biology and ecology of these poorly known reef ecosystems is essential if they are to be included in future spatial management plans and conservation efforts [14]. With that in mind, we ask: (i) is there evidence for distinct benthic communities in MCEs and deeper reefs?; and (ii) to what degree are these deeper communities connected to shallow reefs? Providing thorough baseline information on deeper benthic reef communities is the only way to aid evidence-based decision-making in light of future exploitation of these habitats and their possible role in supporting shallow reef systems.

We focused on Bermuda, an isolated chain of islands situated in western-central Atlantic (figure 1). Despite its temperate latitude, Bermuda has a subtropical climate and is home to the northernmost coral reef systems in the Atlantic [15]. Bermuda's deep reefs are used for a lobster trap fishery and for the capture of deep reef fishes. Thus, it is important to establish a sense of the status and character of the deep reefs to ensure effective resource management in the future. However, even though Bermuda has been a centre for shallow-water research for over a century [16], very little is known from its deeper reef habitats with very limited published quantitative descriptions of changes in the benthic communities within or below MCEs (for a recent review, see [17]). For example, although diverse deep-water taxa, such as corals, are held in museums [18], this information has not been synthesized into community descriptions. Here, we used the latest underwater technology (technical divers with closed-circuit rebreathers and manned submersibles) to investigate reef habitats from 15 to 300 m. We then compared occurrence patterns, diversity and abundance of the whole benthic community and of each component separately, against depth, substratum composition and other measured environmental parameters (conductivity, temperature and salinity).

# 2. Material and methods

## 2.1. Data collection

Four locations were investigated around Bermuda, three on the slope (North Northeast, Spittal and Tiger) and one on the flank of a seamount, Plantagenet Bank (locally known as Argus) (figure 1) during 17 July–14 August 2016 onboard the *R/V Baseline Explorer*. Detailed information on survey design and equipment used can be found in electronic supplementary material, S1. Briefly, reefs between approximately 15 and 90 m were surveyed by technical divers equipped with closed-circuit rebreathers, and using a diver-operated stereo-video system (DOV) consisting of two cameras and lights. Transects followed a 50 m survey tape, about 1.5 m off the bottom, and were approximately 6 min long. Deeper reef locations (approx. 150–300 m) were explored by manned submersibles equipped with a downward-pointing camera with lasers and lights. Transects were approximately 20 min long and covered an estimated distance of 100 m. Characteristics of each transect survey can be found in electronic supplementary

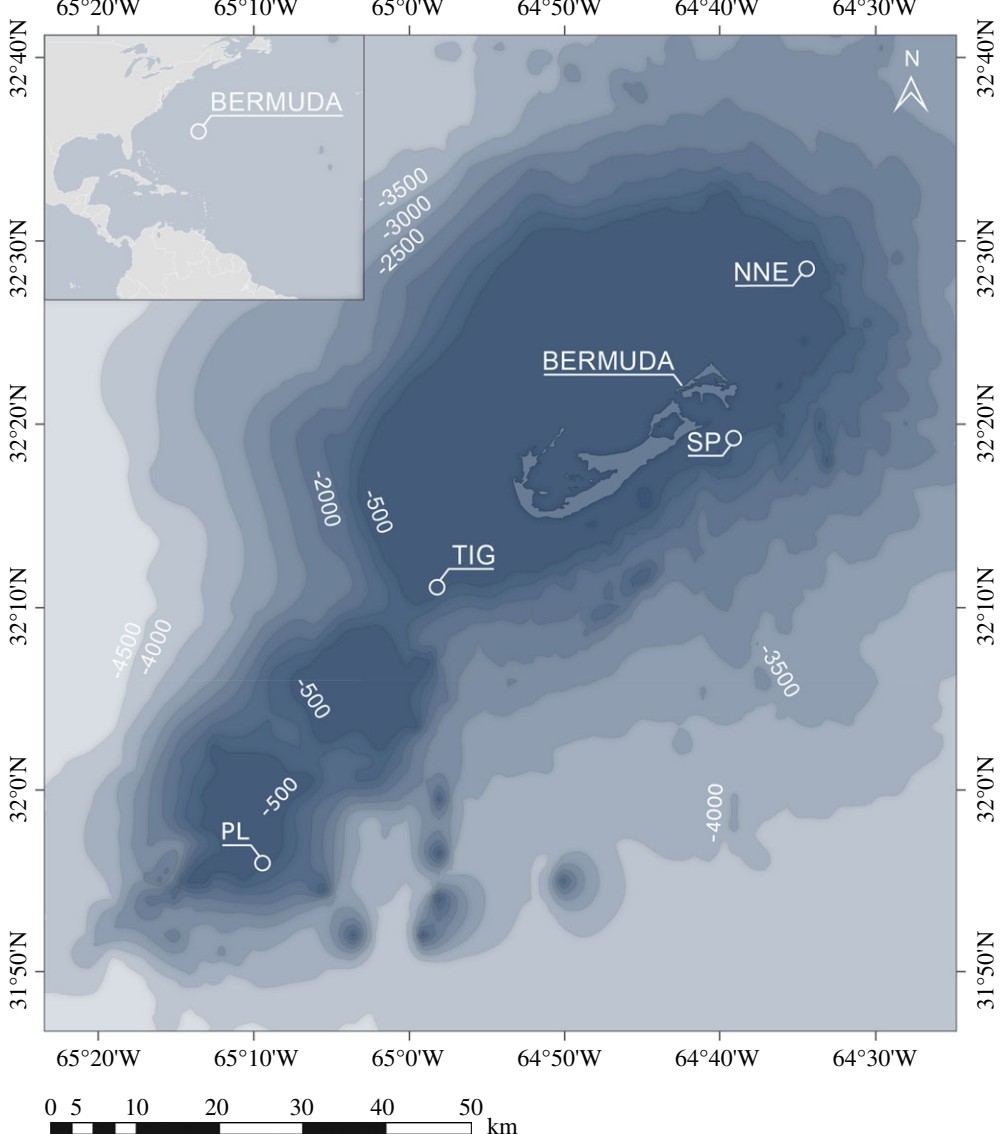

**Figure 1.** Map of study sites around Bermuda. Labels represent depths in metres and were taken from the 2014 GEBCO database. NNE: North Northeast; PL: Plantagenet Bank; SP: Spittal; TIG: Tiger.

material, table S1. Additional environmental parameters such as conductivity, temperature and salinity were also measured across the investigated depths.

## 2.2. Image annotation

Using the retrieved footage from the DOV (note only the footage from one of the two cameras comprising the DOV was used for the benthic analysis) and the downward-pointing camera of the submersibles, still images (JPEGS) were extracted in Adobe Photoshop CC 2015 at a rate of one frame per 10 or 30 s (for data collected with technical divers and submersibles, respectively) in order to avoid spatial overlap. This yielded a total of 3973 frames from 104 transects. Epibenthos quantification was carried out for images that met the following criteria: (i) the image was clear and in focus, (ii) the image was not obscured by sediment resuspension, (iii) both laser dots were clearly visible, and (iv) the area of the image was between 0.6 and 8 m$^2$ for the submersible data and between 0.5 and 8 m$^2$ for the technical diver data (to avoid selection bias for smaller or larger species visible on images very close to or very far from the seafloor, respectively). This led to a subset of 3004 (approx. 76%) images that were subsequently colour-corrected, scaled and analysed in Fiji software [19]. In order to standardize the area per image investigated, we enumerated and identified megafauna within a digital overlay 'quadrat' placed at the centre of the image (diver data: quadrat size 0.5 and 2 m$^2$, for frames between

0.5 and 2, and between 2 and 8 m$^2$, respectively; submersible data: size 0.6 and 2.4 m$^2$ for frames between 0.6 and 2.4, and between 2.4 and 8 m$^2$, respectively). The different quadrat sizes were used to maximize the area of the image investigated.

All megabenthic organisms, defined as individuals large enough (greater than 1 cm) to be clearly seen in underwater imagery [20], were identified to the lowest practicable taxonomic level. Because confident species-level identifications for benthic organisms often require microscopic examination, the majority of taxa were placed into putative morphotypes (i.e. morphologically similar individuals), which represents standard practice in marine biodiversity studies using seafloor imagery (e.g. [21]). Morphotypes that could not be identified to species but appeared morphologically distinct were assigned a unique informal species name (e.g. *Chrysogorgia* sp.). All individuals were enumerated except for encrusting and stoloniferous forms (e.g. algae, some sponges) that were too problematic to count, in which case presence was noted instead.

Substratum composition of each annotated image involved the identification of the following substratum types: (i) sediment (including mud, sand and rock fragments less than 25 cm), (ii) boulders (defined as rocks greater than or equal to 25 cm), (iii) rhodolith beds (defined as unattached accretions of non-geniculate coralline algae), and (iv) bedrock (including sediment veneer, which applies to bedrock beneath a thin sediment layer as indicated by the presence of sessile biota). Area coverage of each type was visually estimated and given a ranking of 1–5 (modified from Polunin and Roberts [22]), representing 1–10%, 11–40%, 41–60%, 61–90% or 91–100% of the seafloor enclosed within the quadrat, respectively. Visual estimates of habitat complexity or percentage cover of benthos and substratum, although semi-quantitative, have been shown to produce results that are comparable to quantitative techniques [23]. Substratum cover per type was expressed as a mean per transect.

## 2.3. Statistical analysis

To account for the unequal sampling of images across transects, we successively estimated the number of morphotypes per transect each time an image was added. Using one-way analysis of variance (ANOVA) followed by Games–Howell pairwise comparisons, we compared the number of morphotypes against the number of images (diver- and submersible-derived transect data were treated separately) and found that seven submersible-derived transects with less than or equal to 13 frames were significantly different ($p \leq 0.05$) from the rest. These transects were taken into account in order to estimate the total number of morphotypes observed during our survey but were excluded from any subsequent diversity and community composition analyses as they were undersampled.

Changes in abundance (numerated taxa only) and morphotype richness with depth were examined using local polynomial regression (LOESS). Changes in benthic community composition were visualized using principal coordinates (PCO) analysis on Bray–Curtis similarity matrices of two sets of data: (i) presence–absence data for the entire benthic community, and (ii) $\log(x+1)$-transformed abundance data—to reduce the effect of common taxa (square-root and fourth-root transformations produced near-identical results)—for a subset of numerated habitat-forming taxa only (i.e. all corals and some sponges; see electronic supplementary material, data S1). Morphotypes highly correlated with the first two PCO axes (i.e. Pearson's correlation coefficient, $\rho \geq |0.6|$) were overlaid in order to reveal the main morphotypes driving assemblage patterns. These were followed by permutational analysis of variance (PERMANOVA) (two fixed factors: depth and site) to identify significant differences between assemblages. All PERMANOVAs in this study were run for 9999 permutations selecting type III (partial) sums of squares and unrestricted permutation of raw data. Site comparisons only were performed on the following three depth subsets: (i) 15–30 m, (ii) 60–90 m, and (iii) 150–300 m transects, which correspond to the shallow, mesophotic and rariphotic reef zones. We chose to do so as the number of conducted transects per depth at a given site was occasionally less than three (e.g. two transects were conducted in TIG at 250 m depth—see electronic supplementary material, table S1), which did not allow statistical comparisons to take place at each depth. Similarity percentage analysis (SIMPER) was also employed in order to pinpoint the taxa responsible for any observed (dis)similarity patterns (one-way design using depth as the group factor; two-way-crossed design using depth and site). Homogeneity of dispersions (as distances to centroids) against depth or site was tested using permutational analysis of multivariate dispersions (PERMDISP) (run for 9999 permutations). Again, comparisons between sites were performed separately for 15–30 and 60–90 and 150–300 m.

In order to investigate whether substratum structure and depth influence benthic communities, we used distance-based linear models (DISTLM) on Bray–Curtis similarity matrices of presence–absence (entire benthic community) and $\log(x+1)$-transformed data (all corals and numerated sponges). We report the results of the best models only (i.e. those with the highest explanatory power identified using the BEST routine and Akaike's criterion). Adding environmental data (conductivity, temperature and salinity; available only for 63 transects) and running the model for the reduced dataset produced comparable results since all three parameters were highly negatively correlated with depth (Pearson's $\rho = -0.78$ to $-0.85$ in all cases). Consequently, only the DISTLM with substratum is presented here.

Using an analysis of beta diversity, which measures community variation across a gradient, we further examined changes in benthic community structure across depth. For that, Jaccard's beta diversity and its components, turnover and nestedness were calculated using presence–absence data for the entire benthic community and selected benthic components separately. Turnover refers to the replacement of some morphotypes by others and is typically the result of environmental filtering or spatial/topographic constraints. By contrast, nestedness is a type of morphotype richness difference leading to ordered species loss [24].

Because of the high number of transects ($n = 11$) conducted in North Northeast at 200 m compared to all other locations at any given depth (electronic supplementary material, table S1), prior to conducting any multivariate and beta diversity analysis, we chose to randomly exclude 6 out of the 11 transects in question so as not to bias our results.

ANOVA was performed in SPSS v. 24 (IBM Corp., Armonk, NY), LOESS in R v. 3.3.3 [25], Jaccard's beta diversity and components in R using the package *adespatial* [26], and multivariate statistical analyses (PCO, PERMANOVA, PERMDISP, DISTLM) were carried out in PRIMER 7 [27] and the add-on package PERMANOVA+ [28].

# 3. Results

## 3.1. Morphotype richness and abundance

A total of 94 morphotypes (i.e. morphologically similar individuals, see section Image annotation) were identified from 104 transects covering a seabed area of 2938 m$^2$. We inspected morphotype accumulation curves at each surveyed depth and found that most were close to reaching a plateau (electronic supplementary material, figures S1 and S2), thus indicating that benthic communities were adequately characterized at every depth. More than a half of the morphotypes were depth-specialists (observed at one or two depths, $n = 56$), only a fifth were depth-generalists (observed at four or more depths, $n = 19$), of these, three morphotypes were found in all depths (all encrusting forms that probably include several species) (figure 2). Overall, there was a decrease in benthic organism abundance with depth (figure 3a). Similarly, morphotype richness decreased with depth for all groups with the exception of algae, whose communities were most speciose at 60–90 m (figure 3b).

All transect information, and presence–absence or abundance data of each morphotype can be found in electronic supplementary material, table S1 and data S1, respectively. Brief descriptions and photographic evidence for the morphotypes are available within figshare: https://doi.org/10.6084/m9.figshare.7333838.v1 [29].

## 3.2. Community structure

PCO analysis suggested depth to be a major driver of megabenthic assemblage composition (figure 4). In general, communities at 15–30 m were largely characterized by scleractinians and octocorals, and to a lesser extent by other taxa (i.e. sponges, hydrocorals and zoanthids). Communities at 60–90 m were mostly associated with algae and to a lesser extent with black corals, while deeper communities (150–300 m) were typified by the black wire corals *Stichopathes* spp. and the octocoral *Hypnogorgia* sp. In addition to the general patterns observed at each broader depth zone (15–30, 60–90, 150–300 m), SIMPER helped to explain within-zone assemblage differences (electronic supplementary material, data S3). For example, transects at 15–30 m although compositionally fairly similar (24–54% avg. dissimilarity based on both, presence–absence and abundance, datasets) still differed in that sea rods (*Eunicea* spp., *Plexaura homomalla*, *Plexaurella* spp., *Pseudoplexaura* spp.) and sea plumes (*Antillogorgia* spp.) were more frequently encountered at 15 m. Transects at 60–90 m had high community dissimilarity (56–95%) largely driven by the higher abundance of black corals (*Antipathes atlantica* and

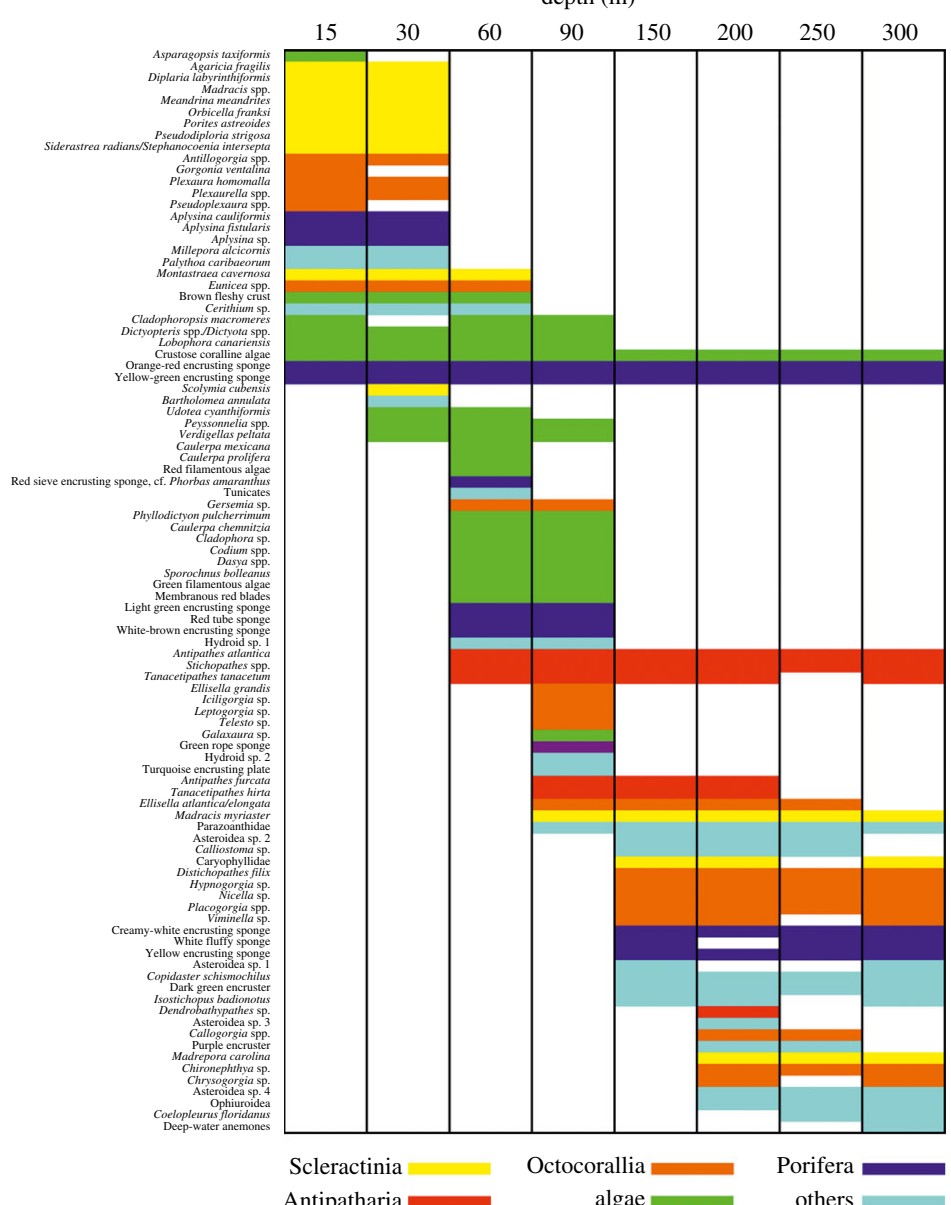

**Figure 2.** Morphotype distribution with depth by category. Coloured and non-coloured boxes indicate presence and absence, respectively.

*Tanacetipathes* spp.), the lower abundance of some algal taxa (e.g. *Caulerpa* spp. and *Lobophora* spp.), and finally the appearance of some octocorals (e.g. *Ellisella* spp.) at 90 m. Differences between the deeper transects were a mixture of subtle differences in the frequency of occurrence of some taxa (e.g. *Hypnogorgia* sp., *Madracis myriaster*) and species replacement (e.g. *Tanacetipathes tanacetum* disappears below 150 m, ophiuroids—typically found as commensals on corals—appear at 200 m).

PERMANOVA using both datasets demonstrated that depth and site were significant ($p < 0.001$ in all cases) in structuring megabenthic assemblages (electronic supplementary material, table S2). Further pairwise comparisons against depth revealed distinct communities at each of the studied depths ($p < 0.05$ in all cases) with the exception of communities between 200 and 300 m (electronic supplementary material, table S2). PERMDISP indicated that there were significant differences in community heterogeneity between investigated depths ($p < 0.001$ for both presence–absence and abundance-based datasets), with transects at 60 and 90 m exhibiting the highest heterogeneity (15 and 30 m: avg. dispersion from centroid 9–16%; 60 and 90 m: 26–60%; 150 and 300 m, 16–30%).

Site-specific comparisons highlighted the unique nature of Plantagenet Bank and North Northeast, both having consistently distinct benthic communities at 60–90 and 150–300 m depths (in both cases

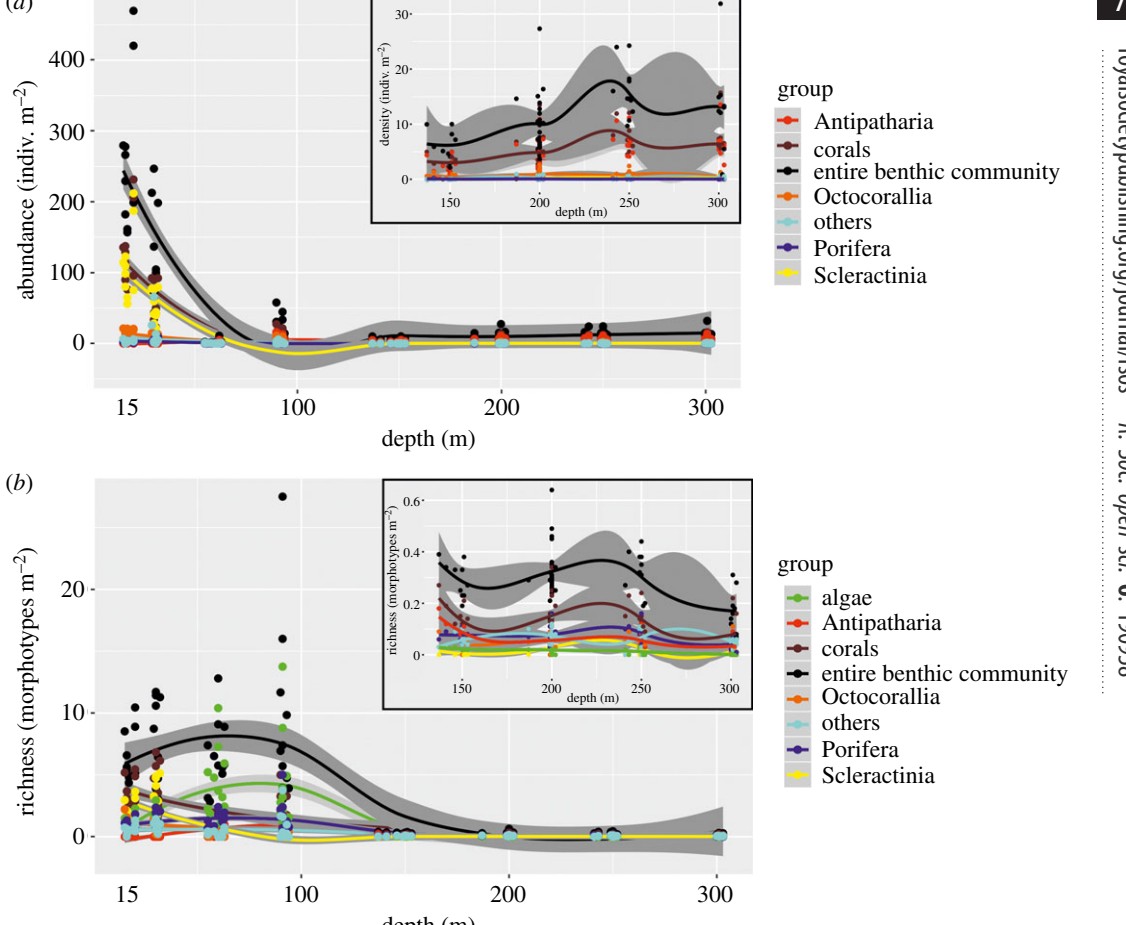

**Figure 3.** Change in abundance (*a*) and morphotype richness (*b*) against depth by category. Both metrics were standardized per m². The insets focus on changes with depth for 150–300 m transects only. Trends are plotted with a LOESS line and their 95% confidence intervals. Corals: Antipatharia + Octocorallia + Scleractinia.

83% of all possible pairwise comparisons against Plantagenet Bank and North Northeast at those depths were significant at the 0.05 level, see electronic supplementary material, table S2). Even though three and five morphotypes were uniquely encountered in Plantagenet Bank and North Northeast respectively, most of the morphotypes were present at all sites. SIMPER results, however, indicated the frequency of occurrence as well as morphotypes abundance rankings changed at these two locations compared to the rest of the sites (electronic supplementary material, data S3).

DISTLM indicated significant relationships between megabenthic community structure and all examined substratum types apart from boulders, irrespective of the type of dataset used (electronic supplementary material, table S3). In all cases, depth accounted for the largest proportion of variance in megabenthic community structure, followed by sediment, bedrock and rhodoliths. All substratum composition and environmental data are available as electronic supplementary material, data S2.

Values of beta diversity with depth, taking into account the entire benthic community, was highest between 30 and 60 m, and between 90 and 150 m, with a smaller peak between 60 and 90 m (figure 5). Beta diversity varied substantially among taxonomic groups partly reflecting the different taxonomic resolutions per benthic component (i.e. low for sponges, high for corals). All coral groups combined, and octocorals, in particular, mirrored the turnover patterns of the entire benthic community being the most widespread group across all depths. In most cases, community differentiation was driven largely by turnover, indicating morphotype replacement across depth. Nevertheless, nestedness was occasionally the driving force of diversity variation, such as in the case of scleractinian communities transitioning from the 30 to the 60 m depth mark, where most shallow-water zooxanthellae coral morphotypes started to disappear. Algal communities transitioning from the 90 to the 150 m depth mark were also highly nested as a result of the almost complete disappearance of this group at greater than 90 m (except for crustose coralline algae).

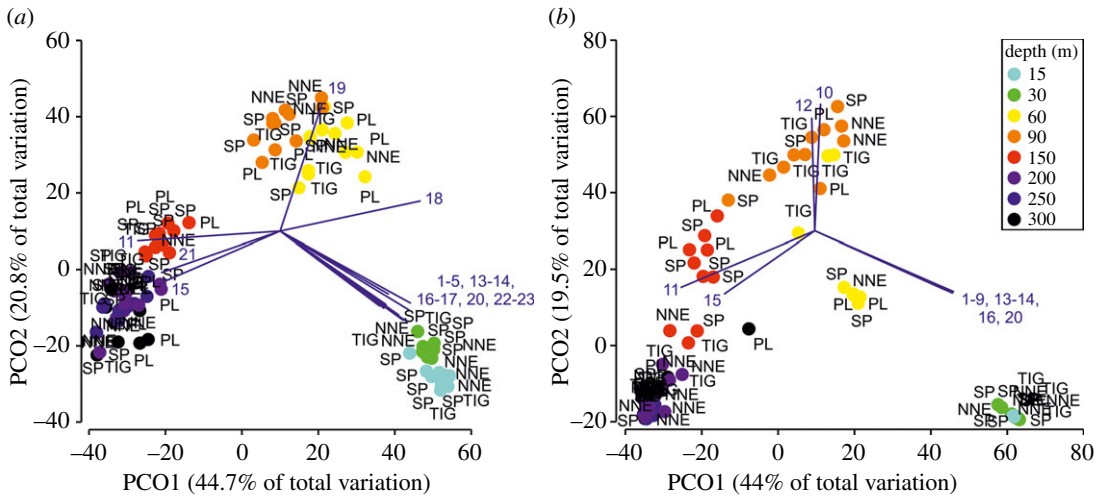

**Figure 4.** PCO analysis ordination of Bermuda's megabenthic communities based on presence–absence (*a*) and abundance data (*b*). Axes describe percentage variation in terms of benthic community structure. Superimposed are vectors of morphotypes highly correlated with the two PCO axes (Pearson's correlation coefficient, $\rho \geq |0.75|$ for (*a*) and $\rho \geq |0.6|$ for (*b*), due to a high number of correlated species in the case of (*a*). The length and direction of the vectors represent the strength and direction of the relationship. Species codes: Scleractinia, 1 = *Agaricia fragilis*, 2 = *Diploria labyrinthiformis*, 3 = *Madracis* spp., 4 = *Meandrina meandrites*, 5 = *Montastrea cavernosa*, 6 = *Orbicella franksi*, 7 = *Porites astreoides*, 8 = *Pseudodiploria strigosa*, 9 = *Siderastrea radians/Stephanocoenia stokesi*. Antipatharia, 10 = *Antipathes atlantica*, 11 = *Stichopathes* spp., 12 = *Tanacetipathes tanacetum*. Octocorallia, 13 = *Antillogorgia* spp., 14 = *Eunicea* spp., 15 = *Hypnogorgia* sp., 16 = *Plexaurella* spp. ALGAE, 17 = Brown fleshy crust, 18 = *Dictyopteris* spp./*Dictyota* spp., 19 = *Sporochnus bolleanus*. Porifera, 20 = *Aplysina fistularis*, 21 = Yellow encrusting sponge. Others, 22 = *Millepora alcicornis*, 23 = *Palythoa caribaeorum*. NNE: North Northeast; PL: Plantagenet Bank; SP: Spittal; TIG: Tiger.

# 4. Discussion

Our analysis reveals strong benthic zonation patterns by depth, identifying unique biological communities at each surveyed depth except between 200 and 300 m (figure 4; electronic supplementary material, table S2). A schematic summary of the main benthic reef zones in Bermuda and their key taxa is presented in figure 6.

## 4.1. Benthic reef zones in Bermuda

We found scleractinian corals decreased rapidly greater than 30 m depth (figure 3). This is similar to the 40 m depth limit that has been reported in previous surveys from Bermuda [30], but in general, much shallower compared to the dense aggregations of hard corals described at greater than 65 m elsewhere [31,32]. Rapidly decreasing levels of irradiance below approximately 40–60 m (electronic supplementary material, figure S3) because of the very steep slopes commencing at those depths around the Bermuda platform (often descending into vertical walls at approximately 100 m) [15] could partly explain the disappearance of this group at depth. Another factor could be high seasonal variation in temperature which has been empirically shown to affect scleractinian recruitment in Bermuda (for a review, see Smith *et al.* [16]). It is likely that low temperatures in the MCEs and deeper reefs in Bermuda, particularly during winter, reduce recruitment and survival rates of scleractinians leading to their low population sizes beyond shallow (less than 30 m) reefs. We also noted a concurrent dramatic increase of algal diversity (figure 3*b*) and cover [33], in particular, that of fleshy macroalgae at MCEs (60–90 m), similar to that reported from the Caribbean [34] and Hawaii [35], which may pre-empt corals in competition for space. A dramatic decline of herbivorous fish abundance and biomass below shallow depths [33] is probably associated with the reduced hard coral cover mentioned earlier. The widespread occurrence of invasive lionfish in Bermuda's MCEs [33,36] may affect juvenile herbivorous fishes at depth [37] and probably reduces grazing pressure on macroalgae allowing them to find refuge in MCEs.

At even deeper depths (150–300 m), we observed a sharp community break at the transition from lower MCEs to deeper reefs, with mainly black corals and to a lesser extent octocorals and other

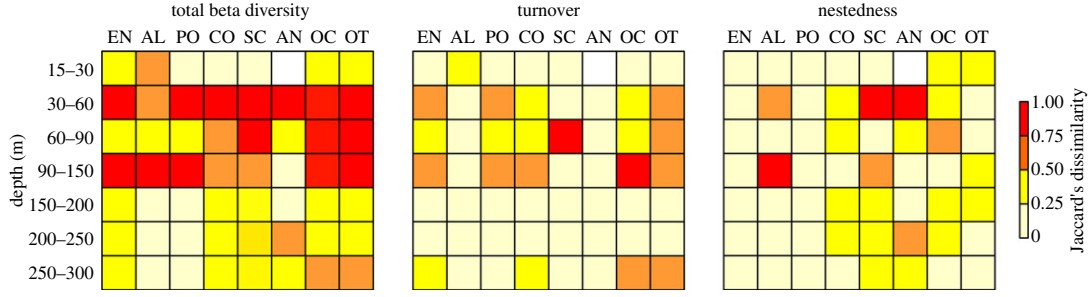

**Figure 5.** Changes in beta diversity and its two components (turnover and nestedness) with depth for the entire benthic community and selected benthic components separately using Jaccard's dissimilarity index. Values close to 0 and 1 indicate low and high dissimilarity, respectively. EN: entire benthic community; AL: algae; PO: Porifera; CO: corals; SC: Scleractinia; AN: Antipatharia; OC: Octocorallia; OT: others.

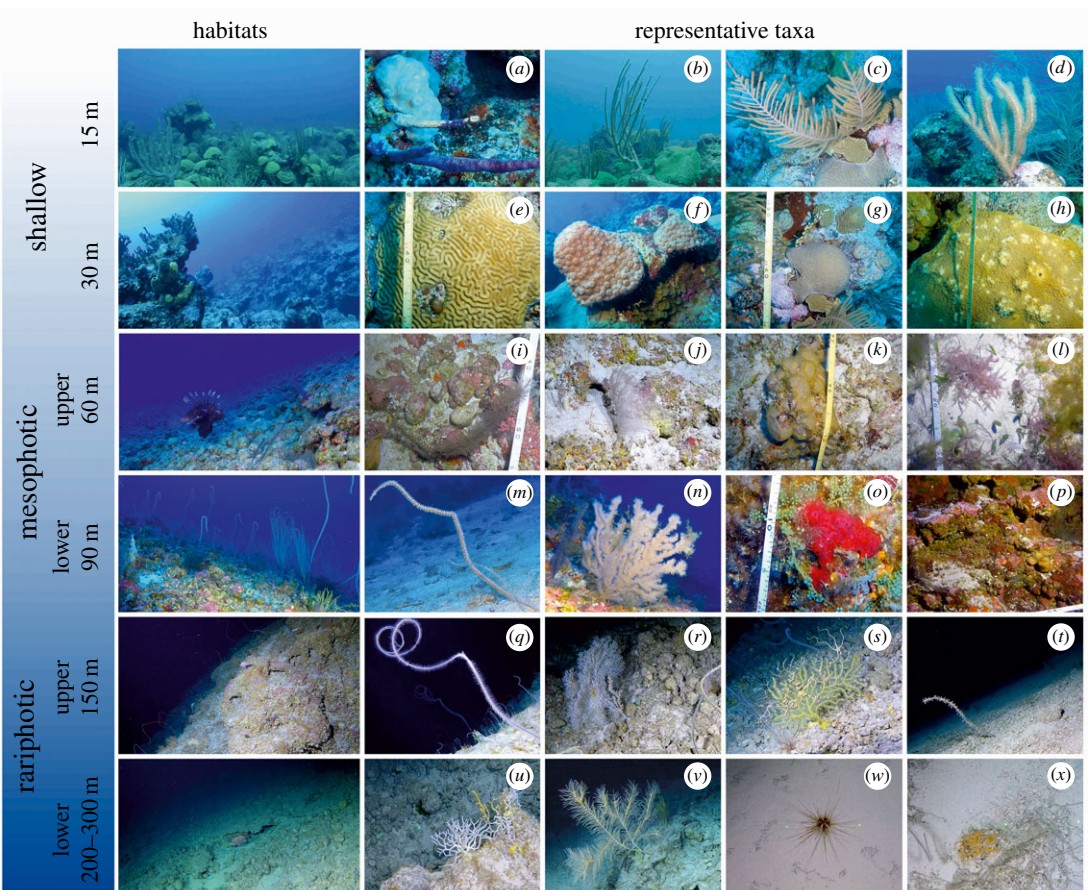

**Figure 6.** Benthic reef zonation in Bermuda with representative taxa. (*a*) *Aplysina cauliformis*, (*b*) *Pseudoplexaura* spp., (*c*) *Antillogorgia* spp., (*d*) *Eunicea* spp., (*e*) *Pseudodiploria strigosa*, (*f*) *Montastraea cavernosa*, (*g*) *Diploria labyrinthiformis*, (*h*) *Orbicella franksi*, (*i*) *Tanacetipathes tanacetum*, (*j*) *Antipathes atlantica*, (*k*) *Lobophora canariensis*, (*i*) *Dasya* spp., (*m*) *Ellisella atlantica/elongata*, (*n*) *Tanacetipathes hirta*, (*o*) red sieve encrusting sponge, cf. *Phorbas amaranthus*, (*p*) various encrusting sponges and crustose coralline algae, (*q*) *Stichopathes* spp., (*r*) *Hypnogorgia* sp., (*s*) *Placogorgia* spp., (*t*) Parazoanthidae, (*u*) *Madracis myriaster*, (*v*) *Callogorgia* spp., (*w*) *Coelopleurus floridanus*, (*x*) Yellow encrusting sponge.

encrusters shared between the two habitats (figures 2 and 5). This deeper community corresponds in depth to the rariphotic zone identified by Baldwin *et al.* [9], and we found some evidence here for differences in communities at 150 m versus 200–300 m depth, possibly corresponding to upper and lower rariphotic zones. Deeper reef habitats were dominated by extensive aggregations of the black wire coral *Stichopathes* spp., similar to those reported at greater than 120 m in Hawaii [35]. We also

observed crustose coralline algae as deep as 301 m close to the deepest ever recorded from Pitcairn Islands (central South Pacific), at 312 m [38].

It is also worth noting the steep decline in benthic organism abundance and morphotype richness with depth (figure 3). The drop in abundance is most likely explained by the concordant decrease in primary production that is reflected in the disappearance of zooxanthellate scleractinian corals and decline of macroalgae cover (figures 2 and 3a; also figure 6 in [33]). The decline in morphotype richness with depth is probably linked to temperature, which was also negatively correlated with depth, as has been previously suggested for tropical ophiuroid diversity in continental shelf (20–200 m) and upper-slope waters (200–2000 m) [39].

Furthermore, location and substratum composition were also found to drive community assembly although to a lesser degree than depth (electronic supplementary material, table S3), with Plantagenet Bank and North Northeast identified as supporting distinct benthic communities from the two remaining sites on the Bermuda platform (Spittal and Tiger) (electronic supplementary material, table S2). The reasons for those differences are best explained by differences in prevailing oceanographic conditions. North Northeast is at the leeward end of the shallow North Lagoon of Bermuda and potentially more affected by the discharge of lagoonal water, with different temperature, salinity and possibly sedimentation conditions [40]. By contrast, Tiger is more upstream from the lagoonal water and thus less influenced from it, while Spittal is completely isolated from the North Lagoon. Owing to its remoteness Plantagenet Bank is unlikely to be influenced by the island of Bermuda apart from fishing impacts. In addition, the complex interplay between currents, topography and food availability, typical of many seamount locations worldwide [41], could also account for the distinct nature of the Plantagenet Bank benthic communities. SIMPER results indicated that these differences were not a result of species replacement since most taxa were found across all studied sites (electronic supplementary material, data S3). This suggests that the distinct nature of Plantagenet Bank and North Northeast communities is not a result of different species being present but instead a result of differences in species abundance and frequency of occurrence (electronic supplementary material, data S3). At least for the Plantagenet Bank seamount, similar results have been obtained for reef fish assemblages as part of the same survey [33] and also correspond to previous comparisons of megabenthic communities from seamounts and adjacent continental margins or slopes (e.g. [42,43]).

## 4.2. Comparisons with other locations

There are few studies that investigate MCE megabenthos at the community level. These have been undertaken in the neighbouring Caribbean (e.g. [34]), Bahamas [44], Indo-Pacific (e.g. [45]) and Australia (e.g. [46,47]). All of them described biodiverse communities harboured within MCEs, as reported here (figure 2). For example, a recent study focusing on reefs in the western Atlantic and Pacific [6] highlighted the unique fish and coral faunas of upper (30–60 m), middle (60–90 m) and lower (90–150 m) MCEs, respectively, which were distinct from those in shallow-water reefs (less than 30 m). Our results of the entire benthos are in close agreement with the above findings further highlighting the biologically unique character of MCEs.

Much less is known from deeper habitats (i.e. approx. 150–300 m), with the exception of three studies around the wider Caribbean region exploring depths between 10 and 309 m [9,44,48]. For example, a study from the Bahamas [44] identified the same broad vertical biological zonation that we report here (i.e. shallow, mesophotic and rariphotic zones); however, community variation was much higher in the case of Bermuda with further sub-divisions within each reef zone (figure 6). Methodological differences between the two studies (e.g. survey design, taxonomic resolution, statistical analysis) could be responsible for these discrepancies. Another study in the Gulf of Mexico [48] was a meta-analysis of field observations, reports and datasets published over a 6-year period, thus, it is not directly comparable with our focused survey around Bermuda. Finally, a Curaçao survey [9] focused on fish assemblages and did not include any benthic data. Nevertheless, for comparative purposes, it would appear likely that the clusters identified here correspond as in Gulf of Mexico [48] and Curaçao [9] to an upper rariphotic zone (150 m, here; 110–200 m, [48]; 130–189 m, [9]), a lower rariphotic zone (200–300 m, here; 190–300 m, [48]; 190–309 m [9]) and an upper and lower mesophotic zone as has been identified elsewhere [49]. Notably, in a parallel study at the same locations in Bermuda, we observed similar trends for reef fish, with unique assemblages found in upper (60 m), lower (90 m) and rariphotic depths (150–300 m), respectively [33].

The zones correspond well in terms of depths but one of the defining features identified for fish communities by Baldwin et al. [9] was that rariphotic taxa (species, genera) belonged to higher

taxonomic groups (e.g. family) that predominantly occur in shallow water. This is much more difficult to establish for the benthos because of the lack of taxonomic resolution. There is no doubt that algae predominantly occur at shallow reefs and MCEs. However, black corals, which were prevalent between 150 and 300 m are distributed over a large depth range, from 2 to greater than 8600 m [50]. In Bermuda, their shallowest presence was in the upper MCEs at 60 m; however, they can be dominant taxa in MCEs in tropical and subtropical regions across the world, including in the Caribbean Sea and Pacific Ocean [51–53]. A higher number of octocoral genera belonging to families typically not encountered in the shallows are found at 150–300 m in Bermuda (e.g. Chrysogorgiidae. Ellisellidae, Nephtheidae, Nidalliidae, Paramuriceidae and Primnoidae), although at least some (e.g. *Hypnogorgia*) belong to families considered to be shallow-water specialists (Plexauridae) (see [18] for more details). *Isostichopus badionotus* is a shallow-water species of holothurian [54]; however, *Coelopleurus floridanus* is found in deep-sea ecosystems greater than 300 m deep [55]. It would seem, therefore, that at the present resolution, the benthos from 150 to 300 m shows a mix of taxa of shallow-water and deep-water origin, although the relative proportions of these groups cannot be resolved at the present time. Further studies at other locations, with a focus on entire communities rather than single benthic components, will be necessary in order to confirm if these patterns are universal or exclusive to Bermuda.

## 4.3. Implications for ocean management

Beta diversity analysis demonstrated that most of the community turnover takes place at the boundary depths of MCEs (shallow to upper mesophotic zone, 30–60 m; lower mesophotic to rariphotic zone, 90–150 m, figure 5), highlighting the ecologically distinct benthic community residing in this zone. Previous studies have reported much higher community overlap between shallow reefs and MCEs (e.g. [56,57]). However, these analyses were typically incorporating historical records dating back years to decades, without taking into account temporal trends in environmental conditions that might have led to shifts in species' distributions (for a recent review, see [3]). Our study suggests that connectivity between the two zones is limited. Furthermore, whatever faunal overlap occurs between these two zones does not necessarily translate into genetic connectivity, and hence the possibility of reseeding potential in the case of a disturbance event. For example, recent studies examining the population genetic structure of selected scleractinian corals from shallow reefs and MCEs (including Bermuda) has been contradictory [58]. Taken together, this suggests that the applicability of the DRRH in Bermuda is more likely to be limited and only applicable to a subset of the total shallow species pool, namely a few eurybathic or depth generalist species for which further population dynamic studies are required to confirm their genetic connectivity through depth. Considering the increasing number of stressors on MCEs and deeper reefs in the rariphotic zone (intense fishing, coral bleaching, invasive species and pollution [6,12,13]) and the unique nature of them, we suggest that these zones should be specifically managed in terms of exploitation activities and for conservation purposes.

Ethics. This study conducted video observations of benthic communities on reefs around Bermuda. No ethical permission was required to undertake this work.

Permission to carry out fieldwork. Research permits for Bermuda were issued by the Department of Environment and Natural Resources, Bermuda (Ship 100 approval no. 87/2016).

Data accessibility. The datasets supporting this article have been uploaded as part of the electronic supplementary material.

Authors' contributions. A.D.R. and L.C.W. conceived and designed the study. A.D.R. directed submersible and technical diving operations. P.V.S. collected the data from imagery with the help of M.R. and H.F., carried out the statistical analysis and prepared the tables and figures. All authors helped with the taxonomic ID of the fauna. The manuscript was drafted by P.V.S. with the assistance of L.C.W. and A.D.R. All authors reviewed the manuscript and gave final approval for publication.

Competing interests. We declare we have no competing interests.

Funding. Nekton received support from the XL Catlin and the Garfield Western Foundation. This project has received funding from the European Union's Horizon 2020 research and innovation programme under grant agreement no. 678760 (ATLAS).

Acknowledgements. This research was undertaken as part of the XL Catlin Deep Ocean Survey Nekton's Mission to the Northwest Atlantic and Bermuda. This output reflects only the author's view and the European Union cannot be held responsible for any use that may be made of the information contained therein. The funders had no role in study design, data collection and analysis, decision to publish or preparation of the manuscript. We wish to thank J. Pitt and T. Warren (Bermudian Government), and G. Goodbody-Gringley and C. Flook (Bermuda Institute of Ocean Sciences; BIOS), for their assistance, advice and participation in the XL Catlin Deep Ocean Survey Bermuda Mission; H. Hirsch (Stanford University) for providing the CTD data; M.C. Stinchcombe (National Oceanography

Centre, Southampton) and M. Taylor (University of Essex) for early discussions. We also wish to thank the following individuals for their contribution in taxonomic identification: G. Goodbody-Gringley (BIOS) and J. Laverick (University of Oxford) (Scleractinia), V. Lovenburg (University of Oxford) (Octocorallia), J. Xavier (University of Bergen) (Porifera), C. Mah (Smithsonian National Museum of Natural History) (Asteroidea). Finally, we would also like to thank the crew and technicians of the *Baseline Explorer*, Brownies Global Logistics and Triton Submersibles and the technical divers of Global Underwater Explorers. This is Nekton Contribution No. 13. Any use of trade, product or firm names is for descriptive purposes only and does not imply endorsement by the US Government.

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
