## [Reviewer comments · Royal Society Open Science]

Review History

RSOS-190958.R0 (Original submission)

Review form: Reviewer 1

Is the manuscript scientifically sound in its present form?

Yes

Are the interpretations and conclusions justified by the results?

Yes

Is the language acceptable?

Yes

Do you have any ethical concerns with this paper?

No

Have you any concerns about statistical analyses in this paper?

No

Recommendation?

Accept with minor revision (please list in comments)

Comments to the Author(s)

General Comment:

Prior to publication, the authors should review relevant chapters in Loya, Puglise and Bridge (2019):

<https://www.springer.com/gp/book/9783319927343>

In particular, Chapter 2 (Goodbody-Gringley et al.) summarizes what is known about MCEs in Bermuda.

Detailed Minor Comments:

Lines 68-69: "However, a distinct zonation for benthic communities below MCEs has yet to be investigated." Does this mean that subdivisions within the "rariphotic zone" haven't been investigated? Or that the "rariphotic" zone itself (as a distinct benthic community) hasn't been investigated? In either case, this should be clarified.

Lines 72-73 "...habitats in the mesophotic zone and beyond to act as a refuge..." The term & abbreviation "MCE"/"MCEs" has already been established, so unless this sentence means something different for "mesophotic zone" (not defined here or elsewhere) than "MCE", then it would be better to be consistent and use "MCEs". Also, "and beyond" is a bit ambiguous; I assume "and deeper" is what is meant. Thus, perhaps change to "...MCE habitats and deeper to act as a refuge..."

Line 86: "...lobster trap fishery and for deep reef fishes." I assume this means a fishery for fishes inhabiting deep reefs? Something more explicit would be good, such as "...lobster trap fishery and for the capture of deep reef fishes." (Or something like that.)

Lines 139, 143, 229: the word "specimens" is generally used to refer to organisms that have been collected and preserved. In this context it appears they indicated organisms are in-situ. Thus, the word "individuals" should be used instead of "specimens".

Line 141: "Since confident species-level identifications..." The word "Since" implies the passage of time. In this context, the word "Because" should be used.

Review form: Reviewer 2

Is the manuscript scientifically sound in its present form?

No

Are the interpretations and conclusions justified by the results?

Yes

Is the language acceptable?

Yes

Do you have any ethical concerns with this paper?

No

Have you any concerns about statistical analyses in this paper?

No

Recommendation?

Major revision is needed (please make suggestions in comments)

Comments to the Author(s)

The research characterizes the Bermuda benthic community across 15-300 m which encompasses shallow, mesophotic, and rariphotic ecosystems. The authors found differences and distinct assemblages between each region, expect between the lower extent of the rariphotic zone. The results of this research are framed in the context of the deep reef refugia hypothesis. Since one of the specific goals of this study was to aid in identifying the role of deeper communities supporting shallow reef systems (i.e. DRRH) I was anticipating a much larger discussion as how their findings relate to the DRRH and management implications beyond the few sentences that are currently presented. Otherwise, as a whole this manuscript is extremely well written and I only have a few minor suggestions to help with clarity that I have noted at the bottom.

With that said, I have my reservations in how the authors categorized many of the morphotypes. The two example that immediately caught my attention were "orange-red encrusting sponge" and "yellow-green encrusting sponge", but my reservations applies to many others. To be frank, these descriptions can apply to a variety of species that are ecologically distinct. Does this description mean that each orange-red encrusting sponge was presumed to be the same species for all observations? Was there a concerted effort to be more selective in categorizing tentative species into orange-red encrusting spp. 1, orange-red encrusting spp. 2, and so forth. I recognize that identifying sponges and some other taxa are difficult to identify to lower taxonomical levels but I feel uneasy with the broad characterization.

=====

Fig 3 - how does abundance and richness go below zero for several groups?; The shallowest depth should be listed on the X-axis.

Fig 4 - location codes should be explained in the figure caption; When the markers get crowded I am unable to distinguish the location codes, particularly in the lower right corner of Fig 4B. I recognize there is a lot of data and I am not sure if there is a solution, however, I must ask, can this be cleaned up?

References - species name should be italicized.

Decision letter (RSOS-190958.R0)

26-Jul-2019

Dear Dr Stefanoudis

On behalf of the Editors, I am pleased to inform you that your Manuscript RSOS-190958 entitled "Low connectivity between shallow, mesophotic, and rariphotic zone benthos" has been accepted

for publication in Royal Society Open Science subject to minor revision in accordance with the referee suggestions. Please find the referees' comments at the end of this email.

Although one reviewer suggests major revision, on reading the review these seem to be effectively minor. Overall, this is a well written contribution and is recommended for publication after appropriate attention to the referees' comments - particularly the issue of morphotype usage.

The reviewers and handling editors have recommended publication, but also suggest some minor revisions to your manuscript. Therefore, I invite you to respond to the comments and revise your manuscript.

- Ethics statement

- Data accessibility

If you wish to submit your supporting data or code to Dryad (<http://datadryad.org/>), or modify your current submission to dryad, please use the following link:
<http://datadryad.org/submit?journalID=RSOS&manu=RSOS-190958>

- Competing interests

- Authors' contributions

- Acknowledgements

- Funding statement

Because the schedule for publication is very tight, it is a condition of publication that you submit the revised version of your manuscript before 04-Aug-2019. Please note that the revision deadline will expire at 00.00am on this date. If you do not think you will be able to meet this date please let me know immediately.

on behalf of Professor Rachel Wood (Associate Editor) and Jon Blundy (Subject Editor)
openscience@royalsociety.org

Reviewer comments to Author:
Reviewer: 1

Comments to the Author(s)
General Comment:

Prior to publication, the authors should review relevant chapters in Loya, Puglise and Bridge (2019): <https://www.springer.com/gp/book/9783319927343>
In particular, Chapter 2 (Goodbody-Gringley et al.) summarizes what is known about MCEs in Bermuda.

Detailed Minor Comments:

Lines 68-69: "However, a distinct zonation for benthic communities below MCEs has yet to be investigated." Does this mean that subdivisions within the "rariphotic zone" haven't been investigated? Or that the "rariphotic" zone itself (as a distinct benthic community) hasn't been investigated? In either case, this should be clarified.

Lines 72-73 "...habitats in the mesophotic zone and beyond to act as a refuge..." The term &

abbreviation "MCE"/"MCEs" has already been established, so unless this sentence means something different for "mesophotic zone" (not defined here or elsewhere) than "MCE", then it would be better to be consistent and use "MCEs". Also, "and beyond" is a bit ambiguous; I assume "and deeper" is what is meant. Thus, perhaps change to "...MCE habitats and deeper to act as a refuge..."

Line 86: "...lobster trap fishery and for deep reef fishes." I assume this means a fishery for fishes inhabiting deep reefs? Something more explicit would be good, such as "...lobster trap fishery and for the capture of deep reef fishes." (Or something like that.)

Lines 139, 143, 229: the word "specimens" is generally used to refer to organisms that have been collected and preserved. In this context it appears they indicated organisms are in-situ. Thus, the word "individuals" should be used instead of "specimens".

Line 141: "Since confident species-level identifications..." The word "Since" implies the passage of time. In this context, the word "Because" should be used.

Reviewer: 2

Comments to the Author(s)

The research characterizes the Bermuda benthic community across 15-300 m which encompasses shallow, mesophotic, and rariphotic ecosystems. The authors found differences and distinct assemblages between each region, expect between the lower extent of the rariphotic zone. The results of this research are framed in the context of the deep reef refugia hypothesis. Since one of the specific goals of this study was to aid in identifying the role of deeper communities supporting shallow reef systems (i.e. DRRH) I was anticipating a much larger discussion as how their findings relate to the DRRH and management implications beyond the few sentences that are currently presented. Otherwise, as a whole this manuscript is extremely well written and I only have a few minor suggestions to help with clarity that I have noted at the bottom.

With that said, I have my reservations in how the authors categorized many of the morphotypes. The two example that immediately caught my attention were "orange-red encrusting sponge" and "yellow-green encrusting sponge", but my reservations applies to many others. To be frank, these descriptions can apply to a variety of species that are ecologically distinct. Does this description mean that each orange-red encrusting sponge was presumed to be the same species for all observations? Was there a concerted effort to be more selective in categorizing tentative species into orange-red encrusting spp. 1, orange-red encrusting spp. 2, and so forth. I recognize that identifying sponges and some other taxa are difficult to identify to lower taxonomical levels but I feel uneasy with the broad characterization.

=====

Fig 3 - how does abundance and richness go below zero for several groups?; The shallowest depth should be listed on the X-axis.

Fig 4 - location codes should be explained in the figure caption; When the markers get crowded I am unable to distinguish the location codes, particularly in the lower right corner of Fig 4B. I recognize there is a lot of data and I am not sure if there is a solution, however, I must ask, can this be cleaned up?

References - species name should be italicized.

Author's Response to Decision Letter for (RSOS-190958.R0)

See Appendix A.

Decision letter (RSOS-190958.R1)

16-Aug-2019

Dear Dr Stefanoudis,

I am pleased to inform you that your manuscript entitled "Low connectivity between shallow, mesophotic, and rariphotic zone benthos" is now accepted for publication in Royal Society Open Science.

Best regards,

on behalf of Professor Rachel Wood (Associate Editor) and Jon Blundy (Subject Editor)
openscience@royalsociety.org

Appendix A

Dear Editor,

Thank you for considering our manuscript entitled “Low connectivity between shallow, mesophotic, and rariphotic zone benthos”.

We appreciate the time and effort put in by the two reviewers. Their constructive comments and suggestions have helped enhance the quality of our revised manuscript. Below you can find our responses to the reviewers’ comments, accompanied where necessary by line numbers so as to locate the revised text in the new version of the manuscript.

Note: The line numbers refer to when the document has tracked changes – show all mark up – selected.

Reviewer 1

Comment 1: Prior to publication, the authors should review relevant chapters in Loya, Puglise and Bridge (2019):

<https://www.springer.com/gp/book/9783319927343>

In particular, Chapter 2 (Goodbody-Gringley et al.) summarizes what is known about MCEs in Bermuda.

Response: We have now added the work of Goodbody-Gringley et al. 2019 in the introduction where we refer to previous work in Bermudian MCEs, and removed four older references that are all covered in the recent review suggested by the reviewer (see line 96-97).

We also want to draw to the attention of the reviewer, that one of the authors of the said Bermuda chapter (Robbie S. Smith), is also a co-author of our submitted paper; hence, we are confident we have incorporated all the latest MCE information from Bermuda in our introduction and discussion.

Where applicable, we have cited other chapters in the introduction and discussion sections (see lines 78, 81, and 444-453).

Comment 2: Lines 68-69: "However, a distinct zonation for benthic communities below MCEs has yet to be investigated." Does this mean that subdivisions within the "rariphotic zone" haven't been investigated? Or that the "rariphotic" zone itself (as a distinct benthic community) hasn't been investigated? In either case, this should be clarified.

Response: We have clarified that in lines 72-73.

Comment 3: Lines 72-73 "...habitats in the mesophotic zone and beyond to act as a refuge..." The term & abbreviation "MCE"/"MCEs" has already been established, so unless this sentence means something different for "mesophotic zone" (not defined here or elsewhere) than "MCE", then it would be better to be consistent and use "MCEs". Also, "and beyond" is a bit ambiguous; I assume "and deeper" is what is meant. Thus, perhaps change to "...MCE habitats and deeper to act as a refuge..."

Response: Done (lines 76-78).

Line 86: "...lobster trap fishery and for deep reef fishes." I assume this means a fishery for fishes inhabiting deep reefs? Something more explicit would be good, such as "...lobster trap fishery and for the capture of deep reef fishes." (Or something like that.)

Response: Done (lines 91-92).

Comment 4: Lines 139, 143, 229: the word "specimens" is generally used to refer to organisms that have been collected and preserved. In this context it appears they indicated organisms are in-situ. Thus, the word "individuals" should be used instead of "specimens".

Response: Done (lines 140, 144, 231).

Comment 5: Line 141: "Since confident species-level identifications..." The word "Since" implies the passage of time. In this context, the word "Because" should be used.

Response: Done (line 146).

Reviewer 2

Comment 1: The research characterizes the Bermuda benthic community across 15-300 m which encompasses shallow, mesophotic, and rariphotic ecosystems. The authors found differences and distinct assemblages between each region, expect between the lower extent of the rariphotic zone. The results of this research are framed in the context of the deep reef refugia hypothesis. Since one of the specific goals of this study was to aid in identifying the role of deeper communities supporting shallow reef systems (i.e. DRRH) I was anticipating a much larger discussion as how their findings relate to the DRRH and management implications beyond the few sentences that are currently presented. Otherwise, as a whole this manuscript is extremely well written and I only have a few minor suggestions to help with clarity that I have noted at the bottom.

Response: We have now added text to address that (lines 444-453). We have also removed the term DRRH from the abstract, so as to not give the impression to the readers that this has been the sole aim of the present study, as the focus has also been on examining the structure of the poorly known deep reef communities.

Although our findings are likely to have implications on coral reef management, elaborating on specific management implications is outside of the scope of our manuscript.

Comment 2: With that said, I have my reservations in how the authors categorized many of the morphotypes. The two example that immediately caught my attention were "orange-red encrusting sponge" and "yellow-green encrusting sponge", but my reservations applies to many others. To be frank, these descriptions can apply to a variety of species that are ecologically distinct. Does this description mean that each orange-red encrusting sponge was presumed to be the same species for all observations? Was there a concerted effort to be more selective in categorizing

tentative species into orange-red encrusting spp. 1, orange-red encrusting spp. 2, and so forth. I recognize that identifying sponges and some other taxa are difficult to identify to lower taxonomical levels but I feel uneasy with the broad characterization.

Response: Identifying organisms from underwater footage is problematic, since the resolution and the distance of individuals from the cameras do not often allow for species-level identifications to take place. Hence, genus or family level IDs are very common for corals, while for groups with wide morphological plasticity and/or groups for which positive identification requires microscopic examination, such as sponges and algae, descriptive names such as the ones we provide in our manuscript are given instead.

In our case, we do not believe that orange-red encrusting or yellow-green encrusting sponge represent one species, hence, the usage of the term morphotype (see also lines 240-243 where we mention that). Since we could not confidently distinguish between different species, genera or indeed families, we decided to group them together and use these descriptive names.

This is standard practice across the marine literature when dealing with video surveys, and is the focus of a paper that is currently in review (Howell et al., two of us, Paris Stefanoudis and Lucy Woodall, are involved in this project). Available at:

<https://www.biorxiv.org/content/biorxiv/early/2019/06/17/670786.full.pdf>

In any case, all of the descriptions for our morphotypes are available in a field ID guide that we have produced (available at <https://tinyurl.com/y2qmbky7>; see also mention of said ID in lines 249-250); all of the individuals involved in that field ID (a lot of them co-authors in the present manuscript) are taxonomic experts in different groups, so we believe the IDs we use are the best possible.

Comment 3: Fig 3 – how does abundance and richness go below zero for several groups?

Comment 4: The shallowest depth should be listed on the X-axis.

Response: This is due the type of statistical analysis we chose. The loess method assumes an unbounded distribution, so can the produced curves (Fig. 3) can sometimes go below 0 if you have data near 0.

For the second point, we have now added the shallowest depth on the x-axis of each plot.

Comment 5: Fig 4 – location codes should be explained in the figure caption:

When the markers get crowded I am unable to distinguish the location codes, particularly in the lower right corner of Fig 4B. I recognize there is a lot of data and I am not sure if there is a solution, however, I must ask, can this be cleaned up?

Response: We tried to clear up Fig. 4B, but due to abundance of the surveyed transects there is always going to be some overlap in the final PCO plots.

Comment 6: References – species name should be italicized.

Response: Done.